# Autophagic Survival Precedes Programmed Cell Death in Wheat Seedlings Exposed to Drought Stress

**DOI:** 10.3390/ijms20225777

**Published:** 2019-11-16

**Authors:** Yong-Bo Li, De-Zhou Cui, Xin-Xia Sui, Chen Huang, Cheng-Yan Huang, Qing-Qi Fan, Xiu-Sheng Chu

**Affiliations:** 1Crop Research Institute, Shandong Academy of Agricultural Sciences, Jinan 250100, China; liyongbo@shangdong.cn (Y.-B.L.); cuidezhou@shandong.cn (D.-Z.C.); suixinxia@shandong.cn (X.-X.S.); srhc.good@163.com (C.H.); sdHCY@163.com (C.-Y.H.); 2School of life science, Shandong Normal University, Jinan 250100, China

**Keywords:** wheat seedlings, drought, autophagy, programmed cell death

## Abstract

Although studies have shown the concomitant occurrence of autophagic and programmed cell death (PCD) in plants, the relationship between autophagy and PCD and the factors determining this relationship remain unclear. In this study, seedlings of the wheat cultivar Jimai 22 were used to examine the occurrence of autophagy and PCD during polyethylene glycol (PEG)-8000-induced drought stress. Autophagy and PCD occurred sequentially, with autophagy at a relatively early stage and PCD at a much later stage. These findings suggest that the duration of drought stress determines the occurrence of PCD following autophagy. Furthermore, the addition of 3-methyladenine (3-MA, an autophagy inhibitor) and the knockdown of *autophagy-related gene 6* (*ATG6*) accelerated PEG-8000-induced PCD, respectively, suggesting that inhibition of autophagy also results in PCD under drought stress. Overall, these findings confirm that wheat seedlings undergo autophagic survival under mild drought stress, with subsequent PCD only under severe drought.

## 1. Introduction

Tissue and organ sizes are genetically determined by cell growth, cell division, and cell death, all of which are influenced by environmental factors [1]. Since plants are unable to escape adverse environmental conditions, they have developed systems aimed at maintaining balance between growth and stress responses, allowing optimal growth during stress [2]. Drought seriously threatens plant productivity, and with climate change, this threat is increasing worldwide [3]. Understanding the mechanisms of plant responses to drought is, therefore, important.

Autophagy, an evolutionarily conserved protein degradation process that responds to environmental stress, is one of the drought tolerance mechanisms employed by plants [4]. During autophagy, the cytosol is sequestered into a double-membraned vesicle called an autophagosome, which fuses with the vacuole to degrade the luminal content [5,6]. Autophagy-related proteins 6 and 8 (ATG6 and 8) are two key proteins involved in autophagosome formation [7]. ATG8 conjugates to the lipid phosphatidyl ethanolamine (PE) to form ATG8-PE, inducing autophagosome formation [8]. ATG8-PE is, therefore, used as an indicator of autophagy [9], together with the presence of autophagic vesicles [10]. Programmed cell death (PCD) is a highly complex, genetically-controlled mechanism of cellular suicide [11]. Found in numerous organisms, PCD eliminates unnecessary and damaged cells during development and offers adaptations to abiotic and biotic environmental stresses [11]. Various methods can be used to determine PCD, such as a DNA fragmentation assay, Trypan blue staining, and a terminal deoxynucleotidyl transferase-mediated dUTP-biotin nick end labeling (TUNEL) assay [12]. As a result, both autophagy and PCD have been observed in plants under different environmental stresses and at various developmental stages. For example, tobacco mosaic virus (TMV) infection was found to cause autophagy and PCD in tomato root tissues [13], while plant cell elimination was also found to require both processes [14]. Moreover, during petal senescence, characteristic morphological features of both autophagy and PCD have also been observed [15]. However, despite the concomitant occurrence of autophagy and PCD in plants, the relationship between the two and the factors determining this relationship remain unknown.

In this study, seedlings of wheat cultivar Jimai 22 were used to examine the occurrence of autophagy and PCD during PEG-8000-simulated drought stress. PEG-8000 promoted autophagy in both a time- and concentration-dependent manner, with autophagy preceding PCD. Moreover, the inhibition of autophagy accelerated PCD, while the duration of drought stress determined its occurrence following autophagy. These findings suggest that wheat seedlings undergo autophagic survival during drought stress, with subsequent PCD occurring only under severe drought. Overall, our study provides new insight into the process of drought tolerance in wheat, aiding breeding programs aimed at increased resistance.

## 2. Results

### 2.1. Drought Stress Represses Seedling Growth of Wheat Cultivar Jimai 22

To investigate wheat seedling growth under drought conditions, PEG-8000 was used to induce drought stress. After treatment with 20–40% PEG-8000 for 48 h, a significant decrease in root and leaf size was observed compared with the control (Figure 1A–C). Moreover, this decrease increased with increasing concentration of PEG-8000. Similarly, a significant decrease in both parameters was also observed after treatment with 20% PEG-8000 for 48–72 h, with a greater decrease in growth with increasing duration (Figure 1D–F). These findings suggest that PEG-8000-induced drought stress repressed seedling growth of wheat cultivar Jimai 22 in both a time- and concentration-dependent manner.

### 2.2. Drought Induces an Increase in ATG6 and ATG8 Expression

To determine whether autophagy occurs after inhibition of seedling growth, the mRNA levels of *ATG6* and *ATG8* in the roots and leaves were analyzed. quantitative real-time reverse transcription-PCR (qRT-PCR) analysis revealed upregulation of both *ATG6* and *ATG8* under 20–40% PEG-8000 treatment for 48 h, with an increase in expression with increasing PEG-8000 (Figure 2A–D). The results also showed a significant increase in the mRNA levels of *ATG6* and *ATG8* under 20% PEG-8000 treatment for 48 to 72 h, with an increase in levels with increasing duration (Figure 2E–H). These findings suggest that PEG-8000 also induces an increase in *ATG6* and *ATG8* expression in both a time- and concentration-dependent manner.

### 2.3. Drought Induces Autophagy in Seedling Roots and Leaves

To confirm the occurrence of autophagy in the roots and leaves of wheat seedlings under drought stress, the anti-APG8A polyclonal antibody from *Arabidopsis thaliana* was used to examine autophagic response since the APG8A protein sequence shows a high similarity with that of the ATG8 in wheat (Appendix A). Western blotting showed two protein bands (14 and 12 kDa, respectively) under PEG-8000 treatment; however, the addition of PLD resulted in a decrease in the 12-kDa band and an increase in the 14-kDa band (Figure 3A), suggesting that the anti-APG8A polyclonal antibody recognizes two forms of ATG8 (ATG8 and ATG8-PE). Western blotting also revealed a significant increase in ATG8-PE under 20–40% PEG-8000 for 48 h and under 20% PEG-8000 for 24 to 72 h in both the roots and leaves, increasing formation under increasing concentration and time (Figure 3B–G), since ATG8-PE is considered to be an indicator of autophagy in plants [10]. In addition, transmission electron microscopy (TEM) observation also showed that autophagic bodies increase significantly under PEG-8000 induction (Figure 4). Therefore, these data confirm that drought, simulated by PEG-8000, induces autophagy in a time- and concentration-dependent manner in wheat seedlings.

### 2.4. The Roots and Leaves of Wheat Seedlings Survive, When Autophagy Occurs

To study whether wheat seedlings undergo PCD when autophagy occurs, TUNEL staining and DNA ladder analysis were performed. Results showed that the 20% of the PEG-8000-treated (96 h)-wheat seedlings looked wilting, and the expression of ATG8-PE had a significant increase (Figure 5A,B), which suggested that the autophagy worked. However, DNA ladder detection and TUNEL staining showed that the classical features of PCD (green nuclei stained by TUNEL) were not observed in both the roots and leaves of wheat seedlings (Figure 5C,D). These data suggest that the roots and leaves of wheat seedlings survive when autophagy occurs.

### 2.5. Autophagy Precedes PCD

To determine the point at which cell death occurs, long-term treatment with 20% PEG-8000 was performed. The trypan blue staining and DNA ladder assay revealed that cell death did not present until 144 and 192 h in the leaves and roots, respectively (Figure 6A–C). However, significant autophagy commenced as early as 24 h after induction (as shown in Figure 3D,E,G). These findings suggest that while PEG-8000-induced drought promotes autophagy at a relatively early stage, cell death is not induced until a much later stage; that is, autophagy precedes cell death. Thus, the duration of drought stress determined the occurrence of PCD following autophagy.

### 2.6. Inhibition of Autophagy Leads to Cell Death

To confirm whether autophagy results in cell survival or cell death, 3-MA, a potent inhibitor of phosphoinositide 3-kinase that inhibits the formation of autophagic structures [17], was used to block autophagy. The effect of knocking down *ATG6* expression was also examined. TEM observations also revealed a significant decrease in autophagic vacuoles after 3-MA treatment (Figure 7A,B), while the Western blot assay revealed a great decrease in ATG8-PE (Figure 7C). Moreover, qRT-PCR showed a decrease in the expression of autophagy-related genes (*ATG3*, *4*, *6*, *7*, *8*, *9*, *10*, *12*, *13*, and *16*) following the addition of 3-MA (Figure 7D). Western blot showed that ATG8-PE had a significant decrease after the knockdown of *ATG6* (Figure 7E–G). These findings confirm that the addition of 3-MA and the knocking down of *ATG6,* respectively, inhibit PEG-8000-induced autophagy. Wilting was observed compared with the control following the addition of 5 mM 3-MA for 72 h and the knockdown of *ATG6* (Figure 8A,C). Trypan blue staining and a DNA ladder assay of the leaves also revealed cell death following the addition of 3-MA and the knocking down of *ATG6* (Figure 8B–E). Taken together, these findings suggest that the inhibition of PEG-8000-induced-autophagy leads to cell death in wheat seedlings, with autophagy acting as a survival process.

## 3. Discussion

Drought severely inhibits seedling growth [18], often resulting in autophagy and PCD [4]. However, the relationship between autophagy and PCD remains unclear, as do the factors determining this relationship under drought stress. In this study, we revealed that autophagy precedes PCD in wheat seedlings during drought stress, with the duration of drought determining this relationship. These findings also revealed that wheat seedlings undergo autophagic survival, with subsequent PCD occurring at a much later stage during more severe drought.

Drought caused by rapid global warming directly impacts agricultural productivity and poses a major challenge to present-day agriculture [19]. Wheat is one of the most important crops worldwide [20], and in arid and semiarid areas, drought stress is the principal abiotic factor affecting wheat yield [21]. Therefore, it is significant to study the regulatory mechanism of wheat on drought stress for improving drought resistance. Our study found that autophagy precedes PCD, and the inhibition of autophagy accelerated PCD under drought stress, which may improve wheat drought stress by enhancing autophagy ability appropriately.

### 3.1. PEG-8000 Induces Both Autophagy and PCD in Wheat Seedlings

In *Triticum dicoccoides*, ATG8 protein and autophagy bodies are abundant in both the leaves and roots during drought stress [22]. Moreover, in common wheat, drought was found to increase the transcript levels of *autophagy-related genes* (*ATGs*) and induce certain proteins involved in the autophagy degradation pathway [23]. Drought stress was also found to induce PCD at a relatively early stage in the endosperm of winter wheat [24]. Therefore, both autophagy and PCD are induced by drought stress in wheat. In our study, using the common wheat cultivar Jimai 22, drought induced an increase in mRNA levels of *ATG6* and *ATG8*, the formation of ATG8-PE, and the number of autophagic vacuoles in both a time- and concentration-dependent manner and also induced PCD in time-dependent manner. Overall, we are the first to establish a molecular model of drought-induced autophagy and PCD in wheat seedlings, which will provide a standard for further study on the mechanisms of autophagy and PCD under drought stress.

Additionally, we use PEG-8000 simulated drought, and in most experiments, 20% concentration was selected because many existing methods for managing substrate water potential have employed PEG to simulate a specific water deficit exposure [25,26]; in wheat seedlings, concentrations of 20% PEG are often selected for related experiments [27,28,29]. Moreover, in our study, 20% PEG-8000 promoted autophagy later than 24 h. Therefore, wheat seedlings were not killed in a short time. The physiological responses to soil drought and PEG-induced osmotic stress are similar in wheat, including decreased leaf relative water contents, water potential, increased osmotic regulation, and enhanced antioxidant capacity [30,31,32].

The mechanism of drought-induced autophagy and PCD may be related to the ABA signal pathway. Abscisic acid (ABA) is a plant hormone involved in stress responses, increasing significantly under drought stress [33]. Under stress, ABA triggers the PYL (ABA receptor)-mediated activation of sucrose non-fermenting-related protein kinase 2 (SnRK2s), which phosphorylates the Target of Rapamycin (TOR) regulator Raptor, thereby inhibiting TOR activity [34]. As a result, inhibition of TOR activity promotes autophagy [35]. Moreover, ABA also induces ROS and H_2_O_2_ accumulation, contributing to plant PCD [36]. However, drought is also thought to induce autophagy via an ABA-independent pathway, for example, by binding the transcription factor ethylene response factor 5 (ERF5) to the promoters of ATG8d and ATG18h, or the heatshock transcription factor A1a (HsfA1a) to the promoters of ATG10 and ATG18f [13,34].

### 3.2. Autophagy Precedes PCD

In plants, autophagy is required for drought tolerance [4], while under severe water stress, the modification of root system architecture occurs due to PCD in the root meristem [37]. Understanding the relationship between PCD and autophagy during drought stress is, therefore, important in crops such as wheat. Although studies have shown the concomitant occurrence of autophagy and PCD in plants [38], whether or not these two processes occur simultaneously remains unknown. In our study, we revealed for the first time that autophagy and PCD do not occur simultaneously in wheat seedlings under drought stress. That is, while autophagy occurred at a relatively early stage, PCD occurred much later, with autophagy preceding PCD. The duration of drought stress was also found to determine occurrence, with short-term drought promoting autophagy and long-term drought promoting PCD. Prolonged drought stress is often associated with ROS accumulation, since decreasing CO_2_ fixation is concomitant with increased electron leakage of triplet oxygen, which may eventually lead to PCD [39]. Moderate drought may produce low levels of ROS. Lower levels of ROS signal positively induce autophagy activity, whereas higher ROS levels lead to rapid PCD [40]. In animals, the mechanism of autophagy and PCD also occurs sequentially, with the cleavage of ATG5 and ATG6 proteins mediating autophagy transformation to apoptosis [41,42]. However, why autophagy precedes PCD in plants under drought stress remains unknown and requires further research in the future.

Based on above study, we can surmise whether PCD or autophagy caused by abiotic stress may be related to the degree and duration time of stress. Moderate or short time stress may result in autophagy, and strong or long-term stress may result in PCD. The degree and duration time of stress may determine the switch from autophagy to PCD.

### 3.3. Autophagy Is a Cell Survival Process under Drought Stress

As mentioned above, drought stress induces autophagy prior to PCD; however, the role of autophagy during drought stress remains unclear in common wheat seedlings. *Atg8*-silenced wild emmer wheat plants were sensitive to drought stress in comparison to the controls [22], while activation of *ATG* genes promoted drought tolerance in tomato [43]. Overexpression of apple ATG18a in tomato and apple plants was also found to increase resistance to drought compared to wild-type plants [44], suggesting that autophagy plays a pro-survival role. Meanwhile, in ipomoea petals, the autophagy inhibitor 3-MA was found to accelerate visual senescence [45], and similarly, our study revealed an increase in drought-induced PCD following 3-MA treatment and the knocking down of *ATG6*, respectively. These findings, therefore, suggest that autophagy also plays a cell survival role in wheat seedlings under drought stress. Regarding the mechanism of autophagy-induced survival under drought stress, some studies suggest that drought stress leads to the accumulation of ROS and H_2_O_2_, which are then possibly removed via the autophagic response, preventing the induction of PCD. Regarding the mechanism of autophagy-induced survival under drought stress, some studies suggest that drought stress leads to an accumulation of ROS and H_2_O_2_, which are then possibly removed via the autophagic response, preventing the induction of PCD [40,46]. However, clarification of this mechanism is required.

## 4. Materials and Methods

Wheat cultivar Jimai 22 originated from the Crop Research Institute, Shandong Academy of Agricultural Sciences. So far, this cultivar has had the largest cultivated area for 6 years in China and also has high yield and stability and is resistant to drought.

### 4.1. Treatment of Wheat Seedlings by PEG-8000

The seeds of wheat cultivar Jimai 22 were germinated at 22 °C. After germination for one week, seedlings of a consistent growth were selected for experiments. These seedlings were firstly incubated in water for 5 days and then treated in two sets of experiments. One was treated with 20% PEG-8000 (P8260, Solarbio Science & Technology Co., Ltd., Beijing, China) for different times (6, 24, 48, 72, 96, 144, 192, 240 h) and another was exposed to a variety of PEG-8000 (10%, 20%, and 40%) for 48 h. Seedlings treated by H_2_O were used as a control. Treatments were performed in a controlled environment chamber under a 16:8 hour light/dark photoperiod and 22 °C temperature conditions. The first leaf about 5 cm from the root tip was sampled during the treatment. The samples were stored in a −70 °C refrigerator for qRT-PCR and Western blot analysis. However, for TEM and PCD detection, fresh samples were used.

### 4.2. qRT-PCR

Total RNA was extracted from the seedlings’ roots and leaves of wheat cultivar Jimai 22 using RNAprep pure plant kit according to the manufacturer’s instructions (DP432, TIANGEN, Beijng, China). After determining the RNA quality by electrophoresis on 1% agarose gel, 2 μg of RNA was reversely transcribed into cDNA through EasyScript One-Step gRNA Removal and cDNA Synthesis Super Mix (L20602, Transgen, Beijing, China). The resulting cDNAs were then used as the templates in PCR reactions. qRT-PCR was performed using TransStart Tip Green qPCR SuperMix (L20803, Transgen, Beijing, China), according to the manufacturer’s instructions and in a real-time thermal cycler (LightCycler^R^ 480II, RhoChe, Basel, Switzerland). *α-Tubulin* was amplified for internal standardization. The experiments were repeated three times, and the experimental data were analyzed statistically using Student’s *t*-test. The relative expression data from the qRT-PCR experiments were calculated using the 2^−ΔΔCT^ method [47]. The primers are listed in Appendix A.

### 4.3. Immunoblotting (Western Blot)

Total proteins from the seedlings’ roots and leaves were extracted using Plant Total Protein Lysis Buffer (P1258, Applygen Technologies Inc, Beijing, China). The protein concentration was measured according to the Bradford method [48]. Equal amounts of protein (30 ug) for each sample were subjected to SDS-PAGE, using a 12.5% gel. Proteins were electrophoretically transferred onto a nitrocellulose membrane. The membrane was incubated with a blocking buffer (2% skim milk powder dissolved in TBS (NaCl, 8.8 g; 2 M, Tris-HCl, pH 7.6, 5 mL and 995 mL H_2_O)) for 1 h at room temperature. The rabbit source polyclonal antibody of *A. thaliana*, APG8A (ab7703, abcam, Cambridge, MA, USA) was diluted to 1:100 in a blocking buffer in TBS and incubated with the membrane overnight at 4 °C. After washing, the membrane was incubated with the secondary antibody (alkaline phosphatase conjugated goat anti-rabbit IgG diluted 1:10,000 in the blocking buffer) (ZB2308, Zhong Shan Jin Qiao, Beijing, China) for 2.5 h at room temperature. The protein signal was visualized using an Alkaline Phosphatase Color Development Kit (C3206, Shang hai, Beyotime, China). The SDS-PAGE gel concentration was 12.5%. The protein bands on the membrane were analyzed using the Quantity One software (http://quantity-one.software.informer.com/).

### 4.4. Assaying the Nature of ATG8-PE by Phospholipase D Cleavage

To confirm that the anti-APG8A polyclonal antibody specially recognizes ATG8 and ATG8-PE proteins, a cabbage phospholipase D (PLD, Sigma, Saint Louis, MO, USA) was used to assay the nature of ATG8-PE [49]. Leaf protein was extracted using Plant Total Protein Lysis Buffer (Applygen Technologies Inc., Beijing, China) and then treated with 20% (g/mL) PEG-8000 for 72 h. Ten units of PLD were then added, and the mixture was incubated at 37 °C for 30 min before boiling for 10 min. Total proteins were separated on 12.5% SDS-PAGE, and then ATG8 was assayed by immunoblotting with the anti-APG8A antibody. Delipidation of ATG8-PE (the membrane-bound form) was recognized as a change in mobility.

### 4.5. Virus Induced Gene Silencing (VIGS) of ATG6

The barley stripe mosaic virus (BSMV)-based VIGS method was used to create gene knockdown plants [50]. A 300 bp fragment of ATG6 from the conserved coding sequence was amplified and purified, with a 300 bp fragment of GFP used as a control. The γ strand of BSMV was then digested in XmacI and fused with the ATG6 or GFP fragment to form the vectors BSMVγ-ATG6 and BSMVγ-GFP, respectively. BSMV-α was linearized with MIuI, BSMV-β was linearized with SpeI, and BSMVγ-ATG6 and BSMVγ-GFP were linearized with BssHII. The linearized vectors were then transcribed in vitro to produce 5ʹ-capped infectious BSMV RNA molecules using the RiboMAX Large Scale RNA Production-T7 Kit (Promega, Madison, WI, USA) with the addition of a cap analog (Promega, Madison, WI, USA) to the transcription mixture. Leaves obtained five days after germination were then mechanically infected with a 1:1:1 mixture of RNAα, RNAβ and RNAγ-ATG6, or RNAγ- GFP in 1 × GKP buffer (50 mM Gly, 30 mM K_2_HPO3, 1% bentonite and 1% kieselguhr). Three replications of each experiment were conducted, with a total of 20 seedlings per each BSMV:ATG6 and BSMV:GFP treatments. After 4 days of BSMV treatment, the seedlings were then treated with 20% (g/mL) PEG-8000 for 72 h. To confirm knockdown efficiency, five plants were randomly selected from each of the two treatments, and the transcript levels of *ATG6* in the first leaves were determined using qRT-PCR.

### 4.6. TEM Analysis

TEM observation was performed to determine autophagic vacuoles [51]. Excised leaf and root samples were immediately cut into small pieces (1 × 2 mm), while minimizing mechanical damage. They were then rapidly fixed in fixation fluid and placed at room temperature for 2 h before being transferred to a refrigerator at 4 °C. The tissue samples were then washed with 0.1 M phosphate-buffered saline (PBS; 140 mM NaCl, 2.7 mM KCl, 10 mM Na_2_HPO_4_, 1.8 mM KH_2_PO_4_, pH 7.4) three times for 15 min each time before being fixed in 1% Osmic acid and 0.1 M phosphate buffer (pH 7.4) at room temperature for 5 h. The samples were then dehydrated in a graded ethanol series (30%, 50%, 70%, 80%, 90%, 95%, and 100%) for 1 h for each concentration and then in a graded mixture of ethanol and acetone (3:1, 1:1, and 1:3) for 0.5 h each, with final dehydration in 100% acetone for 1 h. The samples were then infiltrated in a graded mixture of acetone and Epon 812 (3:1 for 2 h, 1:1 overnight, and 1:3 for 2 h), followed by 100% Epon 812 for 5 h, and then immersed in an embedding plate at 37 °C overnight. The infiltrated tissues were then embedded in Epon 812 at 60 °C for 48 h before slicing into ultrathin (70 nm) sections using an ultramicrotome. Sections were stained with uranyl acetate (2% uranium acetate saturated alcohol solution) and lead citrate for 15 min each, dried at room temperature overnight, and then examined under an electron microscope (HT7700, Hitachi, Tokyo, Japan).

### 4.7. DNA Fragmentation and TUNEL Assay

A DNA fragmentation assay was performed to determine the occurrence of PCD. To do so, 200 mg of root and leaf samples grown under 20% PEG-8000 were harvested at certain time points (48, 96, 144, 192, and 240 h). Isolation of DNA was then performed using an ApopLadder Ex Kit (MK600; Takara Biomedical Technology, Beijing, China) according to the manufacturer’s instructions. To indicate DNA fragmentation, 3 µg of genomic DNA from each sample was used in a standard 100 bp DNA ladder on 1% agarose gel.

Samples were incubated in two changes of pure xylene for 15–20 min each and then dehydrated in two changes of pure ethanol for 10 min each, followed by an ethanol gradient (95, 90, 80, and 70%) for 5 min each before washing in distilled water. The remaining liquid was then removed, and the samples were highlighted using a liquid blocker pen. Proteinase K working solution (20 ug Proteinase K powder distilled in 1 mL PBS) was then added before incubating at 37 °C for 25 min. They were then washed three times with PBS (pH 7.4) in a Rocker device for 5 min each. Excess liquid was then removed, and a permeabilization solution was added before incubating at room temperature for 20 min. The samples were then washed three times with PBS (pH 7.4) in a Rocker device for 5 min each. For the TUNEL reaction, reagent 1 (TdT) and reagent 2 (dUTP) were both from the TUNEL assay kit (Roche, Basel, Switzerland) and were mixed at a ratio of 1:9; then they were placed in a flat wet box with the tissue samples and incubated at 37 °C for 2 h. The box was kept moist by adding water. The samples were then washed three times with PBS (pH 7.4) in a Rocker device for 5 min each. Excess liquid was then removed, and a cover slip was placed over the samples along with anti-fade mounting medium. The slides were then examined using a fluorescent microscope (Axiocam, Carl Zeiss, Germany) with a 590 nm emission wavelength, 400× magnification.

### 4.8. Safranin O and Fast Green (SOFG) Staining

Safranin O and fast green staining were used to indicate wheat cellular morphology. Sections of wheat roots and leaves were incubated in two changes of xylene for 15–20 min each and then dehydrated in two changes of pure ethanol for 10 min each, followed by an ethanol gradient (95, 90, 80, and 70%) for 5 min each before washing in distilled water. For Safranin O staining, sections were immersed in safralin dye for 1–2 h, washed with running water, and then dehydrated in an ethanol gradient (50%, 70%, and 80%) for 3–8 s each. For fast green staining, sections were immersed in fast green dye for 30–60 s, dehydrated in ethanol, and then immersed in xylene for 5 min and sealed with neutral gum. The samples were then examined using a fluorescent microscope (Axiocam, Carl Zeiss, Germany) with a 590 nm emission wavelength, 400× magnification.

### 4.9. Trypan blue staining

Trypan Blue staining is widely used as an indicator of cell death. In this study, leaf samples were immersed in 0.25% (g/mL) Trypan blue solution (0.25 g Trypan blue powder, 12.5 mL phenol, 12.5 mL glycerinum, 48 mL alcohol, 12.5 mL lactic acid, and 14.5 mL distilled water) and then boiled for 2 min before cooling to room temperature. They were then stained for 10 min before destaining with 70% (g/mL) chloral hydrate for three days, refreshing the solution daily. The stained leaves were then observed with a light microscope (Axiocam, Carl Zeiss, Oberkochen, Germany) with 100× magnification.

### 4.10. Quantitative Statistics

The protein bands on the membrane were analyzed using the Quantity One software (http://quantity-one.software.informer.com/). The data of qRT-PCR was analyzed by the graphpad prism 5 software. All experimental data were then analyzed with a Student’s *t* test *, *p* < 0.05; **, *p* < 0.01; ***, *p* < 0.001. Throughout the study, the values are represented as the mean ± standard deviation of 3 independent experiments.

## 5. Conclusions

Based on the results of this study, drought stress leads to the withering of wheat seedlings due to the induction of autophagy at an early stage of drought stress. As drought stress continues, late PCD will then occur, depending on the drought duration. Wheat seedlings, therefore, undergo autophagic survival, with subsequent PCD occurring only during extreme drought stress.

## Figures and Tables

**Figure 1 ijms-20-05777-f001:**
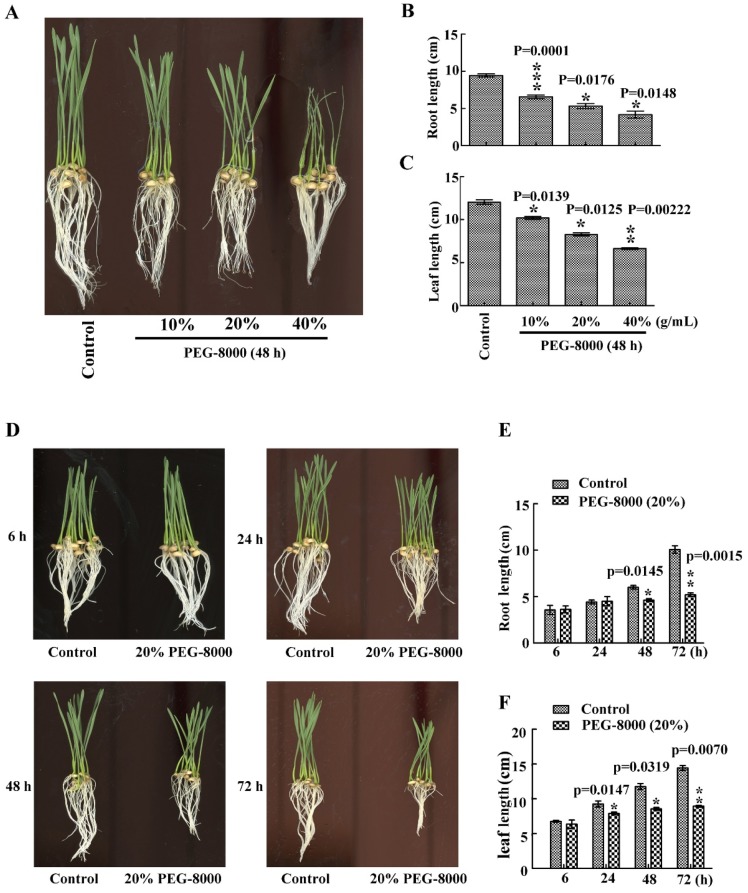
Effect of drought stress on seedling growth of wheat cultivar Jimai 22. Seedling phenotypes (**A**), root length (**B**) and leaf length (**C**) following treatment with 10%, 20%, and 40% PEG-8000 for 48 h. Seedling phenotypes (**D**), root length (**E**), and leaf length (**F**) following treatment with 20% PEG-8000 for 6, 24, 48, and 72 h. Control: treatment with water Bars represent the mean ± SD of three independent experiments. *, **, and *** indicate significant differences at *p* < 0.05, *p* < 0.01, and *p* < 0.001, respectively, according to Student’s *t*-test.

**Figure 2 ijms-20-05777-f002:**
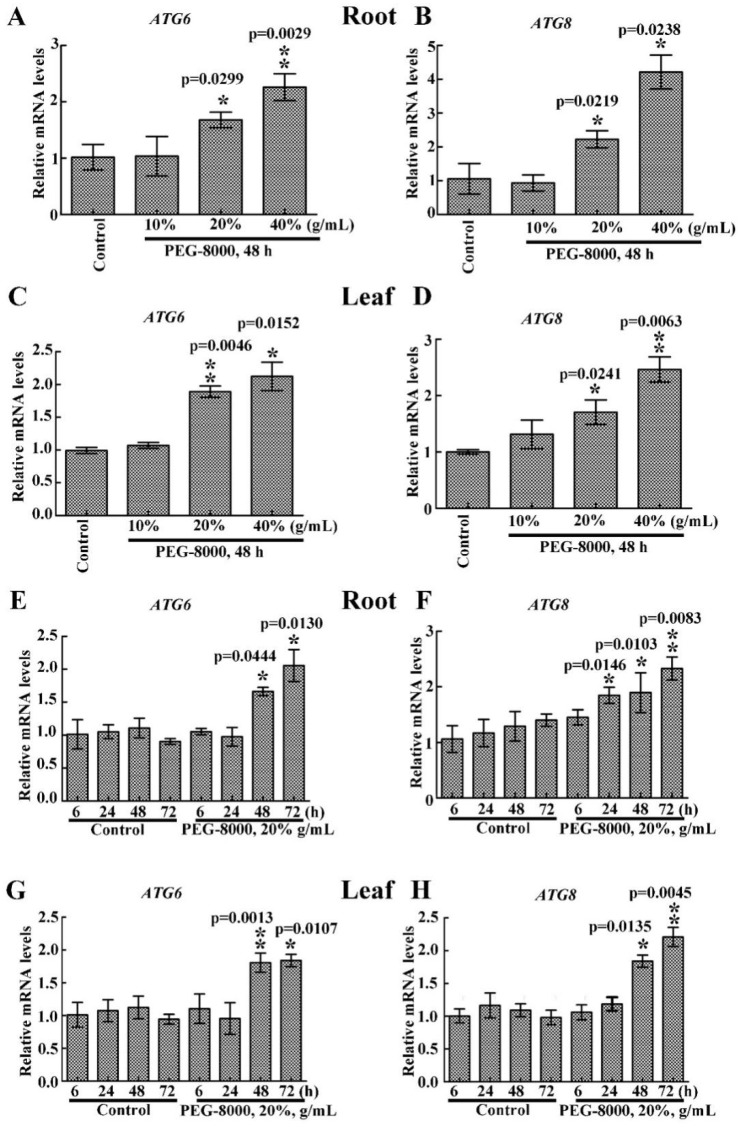
Effect of PEG-8000-induced stress simulating drought on ATG6 and ATG8 expression. effect of 10%, 20%, and 40% PEG-8000 on the root mRNA levels of *ATG6* (**A**) and *ATG8* (**B**) and the leaf mRNA levels of *ATG6* (**C**) and *ATG8* (**D**). Effect of 20% PEG-8000 treatment for 0, 6, 24, 48, and 72 h on the root mRNA levels of *ATG6* (**E**) and *ATG8* (**F**), and leaf mRNA levels of *ATG6* (**G**) and *ATG8* (**H**). α-*Tubulin* was used as an internal reference, and the relative expression was calculated using the 2^−^^ΔΔT^ method. Control: treatment with water. Bars represent the mean ± SD of three independent experiments. * and ** indicate significant differences at *p* < 0.05 and *p* < 0.01, respectively, according to Student’s *t*-test.

**Figure 3 ijms-20-05777-f003:**
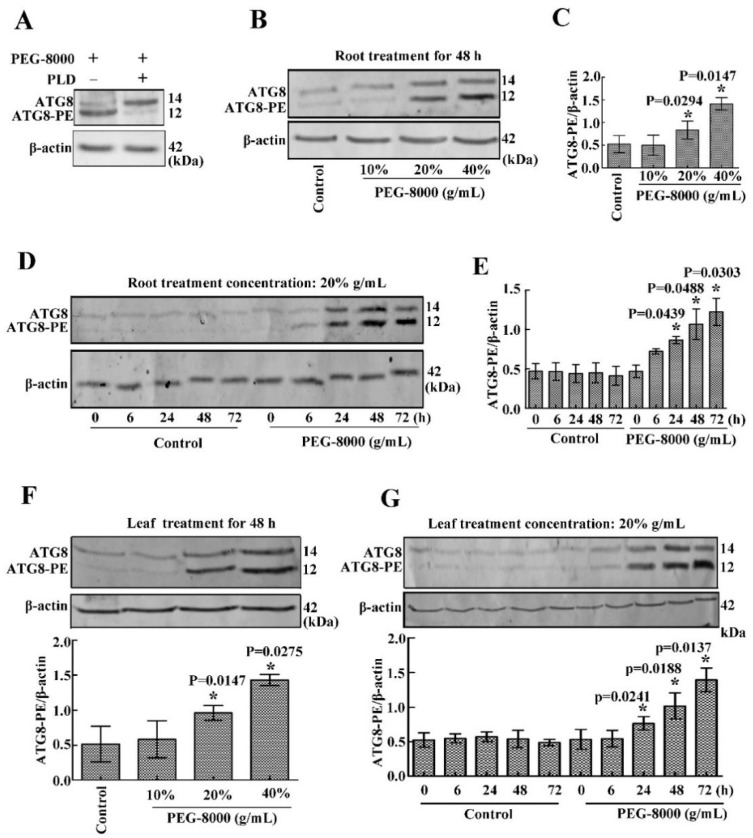
Effect of drought on the induction of autophagy in wheat seedling roots and leaves. (**A**) Effect of phospholipase D on ATG8 and ATG8-PE expression. Treatment with 20% PEG-8000 for 72 h was used as a control (because ATG8 and ATG8-PE protein bands are obvious under this condition). (**B**) Effect of 10%, 20%, and 40% PEG-8000 for 48 h on ATG8 and ATG8-PE expression in the roots. Control: treatment with water. (**C**) Statistical analysis of the results shown in (**B**). (**D**) Effect of 20% PEG-8000 treatment for 6, 24, 48, and 72 h on ATG8 and ATG8-PE expression in the roots. Control: treatment with water. (**E**) Statistical analysis of the results shown in (**D**). (**F**) Effect of 10%, 20%, and 40% PEG-8000 for 48 h on ATG8 and ATG8-PE expression in the leaves. Control: treatment with water. (**G**) Effect of 20% PEG-8000 treatment for 6, 24, 48, and 72 h on ATG8 and ATG8-PE expression in the leaves. Control: treatment with water. β-actin was used as a standard. Data represent the mean ± SD of three independent experiments. * indicates a significant difference at *p* < 0.05 according to Student’s *t*-test.

**Figure 4 ijms-20-05777-f004:**
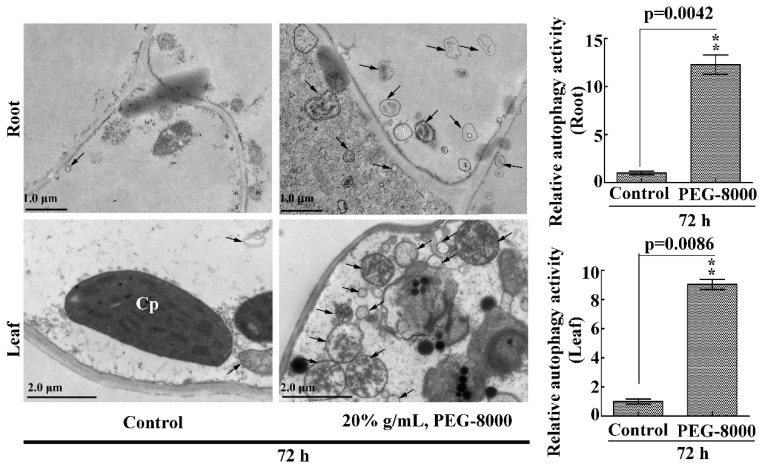
TEM images of autophagic structures in the leaves and roots. Representative ultrastructure of the autophagic bodies (arrows) in the roots and leaves. The relative autophagic activity under 20% PEG-8000 treatment for 72 h was normalized to that of the control plants treated with water only. Approximately 10 cells were used to quantify the autophagic structures under each treatment. Cp, chloroplast; Nu: nuclei. Scale bars = 1 μm (upper images) and 2 μm (lower images). Values in the graphs represent the mean relative activity ± SD of three independent experiments. ** indicates a significant difference at *p* < 0.01 according to Student’s *t*-test.

**Figure 5 ijms-20-05777-f005:**
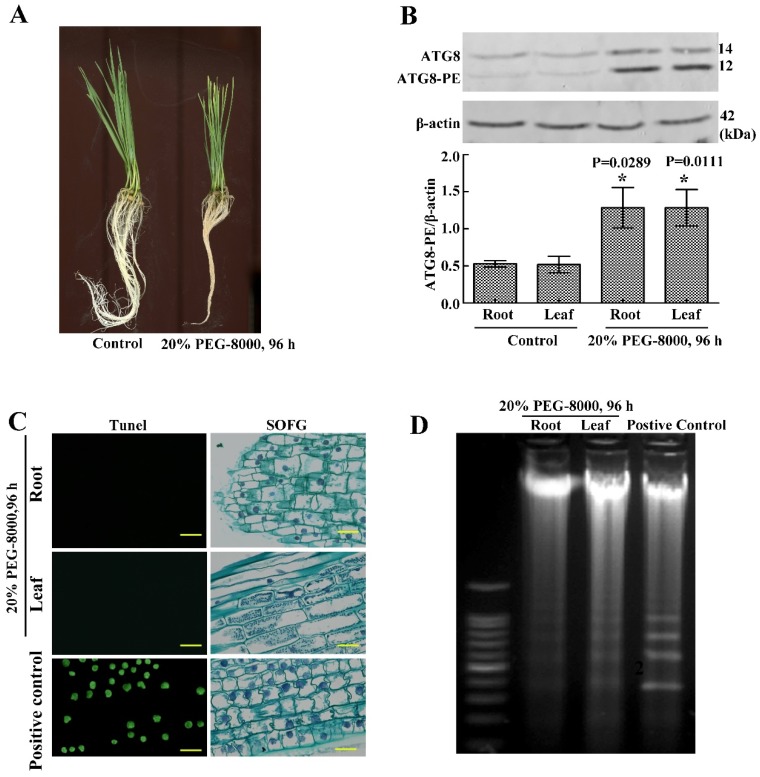
Autophagy as a method of cell survival in the roots and leaves of wheat seedlings following drought stress. (**A**) Phenotypes of wheat seedlings following treatment with 20% PEG-8000 for 96 h. The control was H_2_O. (**B**) Western blotting results showing the effect of treatment with 20% PEG-8000 for 96 h on ATG8 expression. β-actin was used as a standard. Data represent the mean ± SD of three independent experiments. * indicates a significant difference at *p* < 0.05 according to the Student’s *t* test. (**C**) TUNEL and safranin O fast green (SOFG) staining results indicating PCD in the roots and leaves following treatment with 20% PEG-8000 for 96 h. As a positive control, plants were treated with 20% H_2_O_2_ for 96 h (because a high concentration of H_2_O_2_ results in PCD [16], and this concentration is enough to induce PCD). Scale bars = 10 μm. (**D**) DNA ladder analysis of PCD, following treatment with 20% PEG-8000 for 96 h. As a positive control, plants were treated with 20% H_2_O_2_ for 96 h. Data represent the results of three experiments.

**Figure 6 ijms-20-05777-f006:**
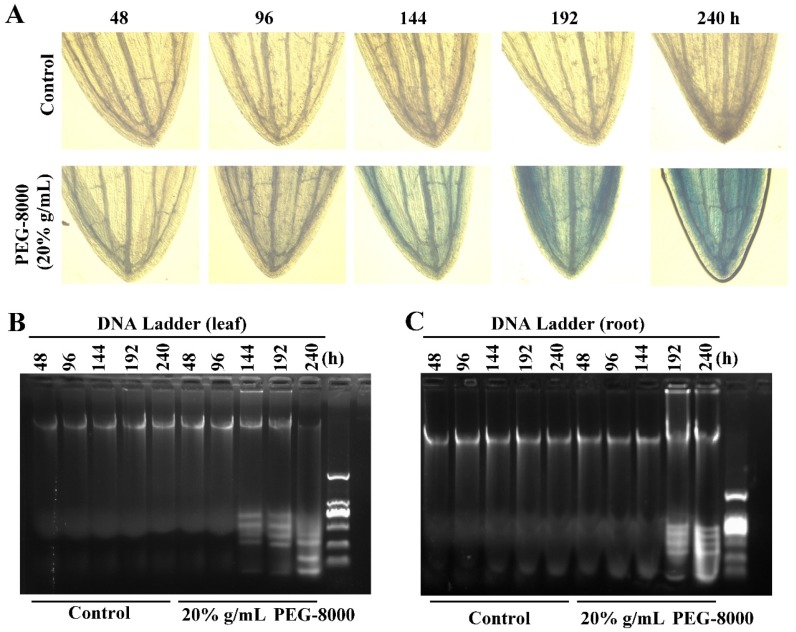
PCD in wheat seedlings following long-term treatment with PEG-8000. (**A**) Trypan blue staining of leaves following treatment with 20% PEG-8000 for 48, 96, 144, 192, and 240 h. (**B**–**C**) DNA ladder assay of PCD in the leaves and roots following long-term treatment with 20% PEG-8000, respectively. Control: treatment with water. Data represent the results of three experiments.

**Figure 7 ijms-20-05777-f007:**
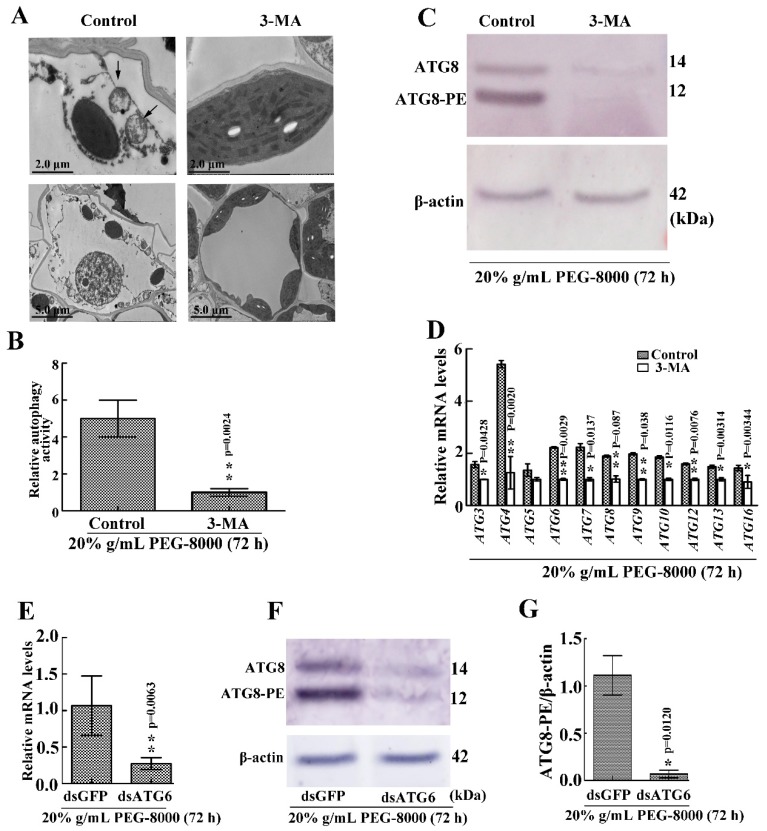
Effect of 3-MA and the knockdown of *ATG6* on PEG-8000-induced autophagy in wheat seedlings. (**A**) Representative ultrastructure of autophagic bodies (arrows) in the leaves following treatment with 5 mM 3-MA in 20% PEG-8000 for 72 h. As a control, water, instead of 3-MA, was used. (**B**) Relative autophagic activity in PEG-8000-treated plants was normalized to that of the 3-MA-treated plants. Approximately 10 cells were used to quantify autophagic structures per treatment. Bars represent the mean ± SD of three independent experiments. ** indicates a significant difference at *p* < 0.01 according to Students *t*-test. Scale bars = 2 μm (upper images) and 5 μm (bottom images). (**C**) Western blot assay of ATG8 protein in the leaves following treatment with 3-MA in 20% PEG-8000 for 72 h. β-actin was used as a standard. (**D**) Leaf mRNA levels of *autophagy-related genes* (*ATG3*, *4*, *5*, *6*, *7*, *8*, *9*, *10*, *12*, *13*, and *16*) following 3-MA treatment. *ɑ-Tubulin* was used as the internal reference, and the relative expression was calculated using the 2^−^^ΔΔT^ method. (**E**) qRT-PCR analysis of the knockdown efficiency of *ATG6*. *α-Tubulin* was used as the internal reference, and the relative expression was calculated using the 2^−^^ΔΔT^ method. (**F**) Level of ATG8-PE in the leaves after *ATG6* knockdown. (**G**) Statistical analysis of the effects of *ATG6* RNAi on ATG8-PE formation. Bars represent the mean ± SD of three independent experiments. * and ** indicate significant differences at *p* < 0.05 and *p* < 0.01, respectively, according to Student’s *t*-test.

**Figure 8 ijms-20-05777-f008:**
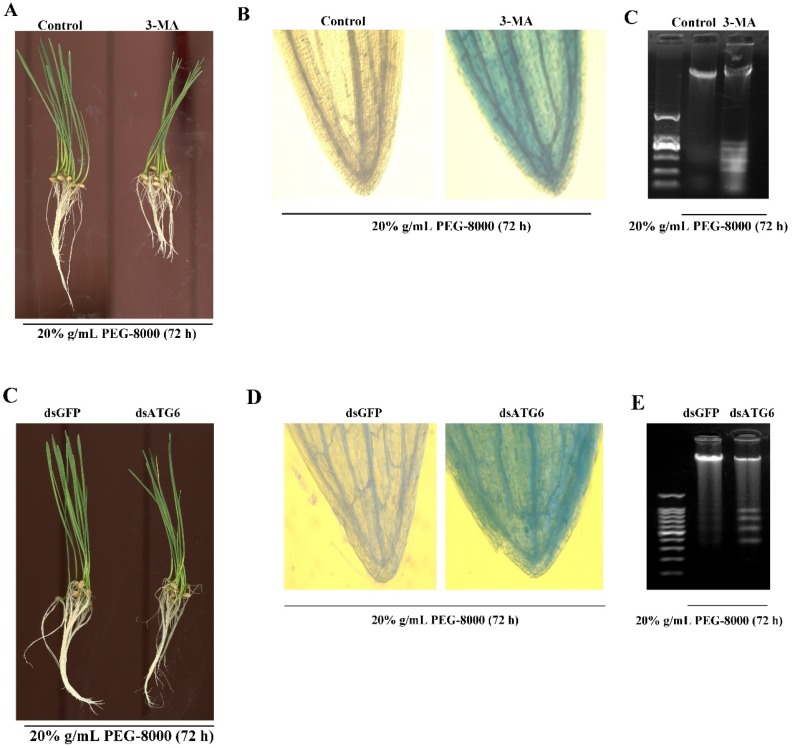
Acceleration of PEG-8000-induced PCD following inhibition of autophagy. (**A**) Seedling phenotypes following treatment with 3-MA in 20% PEG-8000 for 72 h. As a control, water instead of 3-MA was used. (**B**) Trypan blue staining of leaves following 3-MA treatment. (**C**) DNA ladder assay of PCD in the roots following 3-MA treatment. (**D**) Seedling phenotypes following *ATG6* knockdown. dsGFP was used as a control. (**E**) Trypan blue staining of leaves following *ATG6* knockdown. (**F**) DNA ladder assay of PCD in the roots following *ATG6* knockdown. Data represent the results of three experiments.

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
