# Peer review of "Autophagic Survival Precedes Programmed Cell Death in Wheat Seedlings Exposed to Drought Stress"

_ijms, 2019, doi:10.3390/ijms20225777_

Round 1
Reviewer 1 Report
The authors have addressed successfully the concerns risen in the previous revision and now the manuscript can be published.
Author Response
Thank you
Reviewer 2 Report
The authors of this manuscript seek to understand the sequence of autophagy and programmed cell death in s selected wheat cultivar, in response to drought stress, simulated by polyethylene glycol (PEG) 8000 treatment. The authors found that autophagy precedes programmed cell death, occurring in the early stages of simulated drought exposure, while PCD being induced later upon prolonged stress. They also indicate that the inhibition of autophagy results in PCD induction, accelerating cell death.
The study is relevant, contributing new base information towards a better understanding of plant mechanistic responses to drought stress. Nevertheless, the manuscript should be improved prior for consideration for publication. I am listing my observations/questions in the sequence they occur in the text.
The text needs adjustments to be uniform (fonts, bolding, etc.) throughout the manuscript, following the author guidelines. In the title “undergoes” should be replaced with “undergo”, or the title should be reformulated for better flow. One example: Autophagic survival precedes programmed cell death in wheat seedlings exposed to drought stress. Abbreviations should be explained and full terms used where they first occur in the text, including the Abstract. For example line 13 – PEG-8000 or line 17 – ATG 6. Line 19: replace “…autophagic survival during drought stress ….” with “…autophagic survival under mild (or in the early stages of…?) drought stress ….”. Line 51 Include in the Introduction the question this study was aimed to answer, and discuss (either in the Introduction or in the Discussion) how the obtained data will advance base knowledge of plant stress responses and the potential applied relevance of the data, including why is it important to gain a better understanding of plant responses to environmental impacts, such as drought, which could be one of the possible consequences of a warming climate.
Materials and Methods: seems to be missing details on the used wheat cultivar. Where the seedstock originated from? Why was this cultivar selected? Is it a databased and characterized cultivar? For the purpose of the study: is this cultivar drought sensitive or drought resistant?
1.subtitle needs to be re-formulated. One possibility: PEG-8000 treatment of wheat seedlings. No need for the term “chemical”.
2.The entire paragraph under 2.1. in the Material and Methods needs to be reformulated. In its current form it is unclear how the PEG treatments were carried out. Seedlings were treated in two sets of experiments? One with 20% PEG for up to 240 hours, with samplings at different intervals, and in a separate set of experiments only for 48 hours but exposed to a variety of PEG concentrations? Why was the 20% PEG treatment selected for long term response measurements and not the other concentrations? How were roots/leaves sampled at the different intervals during the treatment? Were standardized leaf/root disks or amounts flesh frozen prior to molecular analysis? Were any of the samples stored? If so, under which conditions? Line 63: Instead of “the whole culture process…” use “ treatments were performed in a controlled environment chamber under 16:8 hours light/dark photoperiod and 22 C temperature conditions”. Line 73: RhoChe thermocycler needs specifications for the manufacturer. Line 73: …Tubulin was also amplified… Line 76: The used primers are listed in Table 1. Table 1: would recommend moving the primer sequences into the Supplementary Materials. Also, add the names of the listed genes.
3. Subsection: Expand title from Immunoblotting to explain what was immunoblotted Line 90: “The SDS-PAGE gel concentration was 12.5%” is unnecessary. Include this information where logically belongs to, into the sentence in lines 81-82. For eg: Equal amounts of protein (30 ug) for each sample were subjected to SDS-PAGE, using a 12.5% gel. Explain the TBS abbreviation used here.
4. Subtitle needs to be reformulated. What the authors mean by assaying the nature of…? Line 94: for consistency use 20% PEG only in the text.
5. subsection needs references to the used procedures. Include details on the mechanical infection. How? Reference! 6. Expand subtitle to include what was imaged. Remove from line 116-117 : “The detailed method was a follows” – unnecessary. Reference the TEM method used. Line 120-121: 0.1M phosphate buffer: sodium or potassium phosphate? 7 – line 133: ….”leaf samples grown under certain conditions were harvested at certain timepoints” – specify details instead of using “certain” Lines 135-137 – sentence “To visualize…” needs to be reformulated, is incorrect Line 137 – remove sentence “Tunel staining was carried out…” - unnecessary Line 138 – specify xylene concentration Line 141 – “Proteinase K working solution”. Specify concentration/amounts used or units. Lines 150-151 – Germany – spell with capital letters. Include details on the microscopy observations! Wavelengths, magnification, etc…or reference the used technique. The same in 2.8 and 2.9! Include as 2.10 all details that pertain to Data analysis and statistics used! 3.1. – uses % PEG consistently (not mixed with % g/mL PEG). Remove the word “chemical” from line 176, unnecessary Figure 1 description: consolidate panel descriptions to avoid redundancy. For Eg. Seedling phenotypes (A), root length (B) and leaf length (C) following treatment with 10, 20 and 40% PEG-8000 for 48 hours…..Use this consolidation in all Figure descriptions. Figure 2. 20% PEG treatment at the 48 hours time point in roots on panel B (2.3 or 2.4) and F (1.8?) shows different relative mRNA levels, though those points overlap between approaches. How do the authors explain this discrepancy? If these data derive from different experiments, these differences (variations between individual samples) may be larger than the significant differences and the trend shown on panel F… Line 216 : “Since ATG8-PE is considered…” is unfinished sentence. Modify!
Figure 3, Lines 222-223: “Treatment with 20% PEG-8000 for 72 hours was used as a control”. Explain why!
4. subtitle and sentence in lines 247-248 needs to be reformulated, does not make sense: “….wheat seedlings are survival (survive?) when autophagy occurs” Line 243: Remove “the following experiments were performed”. This is the results section, does not belong here.
Figure 5 presents data from samples collected after 96 hours. Why this time-point and not some of the later or earlier points? The authors need to find a coherent manner to explain why do they present the array of data and why selected to present only some sets of data at a variety of different time-points during treatments. Line 257 “ As a positive control plants were treated with 20% H2O2”. Why? These control treatments and their justification should be included into the appropriate sections of the Materials and Methods.
Figure 6 – Are these panels representative examples of a larger set of data? (mentioned 3 experiments..) Than specify that these are representative examples and not averages.
Discussion: The discussion is very superficial and does not concentrate on the evidence and data the paper presents, aligning data with the state of the art of current knowledge in the field of plant stress responses. The authors spend a disproportionately long section in the discussion in detailing potential mechanisms of drought induced autophagy and PCD, but their experiments did not investigate these mechanisms and did not show data in support of the described mechanisms.
All this, at the expense of neglecting focusing on their own findings. I recommend re-writing and expanding the discussion to address the data specifically presented in the manuscript. Line 323 “In plants” is repeated. Lines 352 and 353 – “however” is used in two consecutive sentences- redundant. Reformulate!
Figure 9 does not add specific value to the manuscript. It is clear already from the data that autophagy precedes PCD and PCD will not occur until the late, sever phases of stress. Either remove Figure 9, or expand it and tie in additional details on the hypothesized molecular / cellular mechanisms and details behind autophagy and PCD based on information available in the literature to integrate the novel information, as also discussed/hypothesized in the Discussion.
Author contributions: “…analyzed the experiments..”, maybe mean “analyzed the data…”?
Author Response
Dear reviewer 2
Thank you very much for pointing out so many mistakes, giving us so many good advice. The answers are as follows:
The authors of this manuscript seek to understand the sequence of autophagy and programmed cell death in s selected wheat cultivar, in response to drought stress, simulated by polyethylene glycol (PEG) 8000 treatment. The authors found that autophagy precedes programmed cell death, occurring in the early stages of simulated drought exposure, while PCD being induced later upon prolonged stress. They also indicate that the inhibition of autophagy results in PCD induction, accelerating cell death.
The study is relevant, contributing new base information towards a better understanding of plant mechanistic responses to drought stress. Nevertheless, the manuscript should be improved prior for consideration for publication. I am listing my observations/questions in the sequence they occur in the text.
The text needs adjustments to be uniform (fonts, bolding, etc.) throughout the manuscript, following the author guidelines.Answer: We have made adjustments to be uniform throughout the manuscript, following the author guidelines.
2.In the title “undergoes” should be replaced with “undergo”, or the title should be reformulated for better flow. One example: Autophagic survival precedes programmed cell death in wheat seedlings exposed to drought stress.
Answer: We have replaced with “undergo” and used the title “Autophagic survival precedes programmed cell death in wheat seedlings exposed to drought stress.”
Abbreviations should be explained and full terms used where they first occur in the text, including the Abstract. For example line 13 – PEG-8000 or line 17 – ATG 6. Line 19: replace “…autophagic survival during drought stress ….” with “…autophagic survival under mild (or in the early stages of…?) drought stress ….”.
Answer: We have explained the abbreviations and full terms used where they first occur throughout the manuscript, and replaced “…autophagic survival during drought stress ….”with “…autophagic survival under mild drought stress.
Line 51 Include in the Introduction the question this study was aimed to answer, and discuss (either in the Introduction or in the Discussion) how the obtained data will advance base knowledge of plant stress responses and the potential applied relevance of the data, including why is it important to gain a better understanding of plant responses to environmental impacts, such as drought, which could be one of the possible consequences of a warming climate.
Answer: We added some description in “Discussion” as follows “Drought caused by rapid global warming directly impacts agricultural productivity and poses a major challenge to the present-day agriculture [1]. Wheat is one of the most important crops worldwide [2], and in arid and semiarid areas, drought stress is the principal abiotic factor affecting wheat yield [3]. Therefore, it is significant to study the regulation mechnism of wheat on drought stress for improving drought resistance. Our study found that autophagy precedes PCD and inhibition of autophagy accelerated PCD under drought stress; which may improve wheat drought stress by enhancing autophagy ability appropriately”
Materials and Methods: seems to be missing details on the used wheat cultivar. Where the seedstock originated from? Why was this cultivar selected? Is it a databased and characterized cultivar? For the purpose of the study: is this cultivar drought sensitive or drought resistant?
Answer: We added some details about wheat cultivar and described like this “wheat cultivar Jimai 22 was originated from Crop Research Institute, Shandong Academy of Agricultural Sciences. So far, this cultivar has the largest cultivated area for 6 years in China and also has high yield and stability, and is resistant to drought.”
subtitle needs to be re-formulated. One possibility: PEG-8000 treatment of wheat seedlings. No need for the term “chemical”.Answer:We have deleted the term “chemical” throughout the manuscript.
The entire paragraph under 2.1. in the Material and Methods needs to be reformulated. In its current form it is unclear how the PEG treatments were carried out. Seedlings were treated in two sets of experiments? One with 20% PEG for up to 240 hours, with samplings at different intervals, and in a separate set of experiments only for 48 hours but exposed to a variety of PEG concentrations?
Answer:We have reformulated the entire paragraph and described like this “The seeds of wheat cultivar Jimai 22 germinated at 22°C. After germination for one week, Pick out seedlings that grow consistently. These seedings were firstly incubated in water for 5 days and then treated in two sets of experiments. One was treated with 20% PEG-8000 (P8260, Solarbio Science & Technology Co., Ltd, Beijing, China) in different time (6, 24, 48, 72, 96, 144, 192, 240 h) and another was exposed to a variety of PEG-8000 (10%, 20%, and 40%) for 48 h.”
Why was the 20% PEG treatment selected for long term response measurements and not the other concentrations?
Answer: We explained it in “discussion” part, as follows: “Additionaly, we use PEG-8000 simulated drought and in most experiments, 20% concentration was selected; because many existing methods for managing substrate water potential have employed PEG to simulate a specific water deficit exposure[4, 5]; and in wheat seedlings, the concentration of 20% PEG are often selected for related experiments [6-8]. Moreover, in our study, 20% PEG-8000 promoted autophagy just over 24h; and under this concentration, wheat seedlings are not be killed in a short time. The physiological responses to soil drought and PEG stress are similar in wheat, including lower leaf relative water content, lower water potential, increased osmotic regulation, and enhanced antioxidant capacity [9-11]”.
”
How were roots/leaves sampled at the different intervals during the treatment? Were standardized leaf/root disks or amounts flesh frozen prior to molecular analysis? Were any of the samples stored? If so, under which conditions?Answer: We added some details in “materials…” part, as follows “The first leaf and about 5 cm from root tip were sampled during the treatment. The samples can be stored in a -70℃ refrigerator for the following Quantitative real-time reverse transcription-PCR (qRT-PCR) and Western blot analysis. However, fresh samples must be provided for Transmission electron microscopy (TEM) and PCD detection.”
Line 63: Instead of “the whole culture process…” use “ treatments were performed in a controlled environment chamber under 16:8 hours light/dark photoperiod and 22 C temperature conditions”. Line 73: RhoChe thermocycler needs specifications for the manufacturer. Line 73: …Tubulin was also amplified… Line 76: The used primers are listed in Table 1. Table 1: would recommend moving the primer sequences into the Supplementary Materials. Also, add the names of the listed genes.Answer: We have used the sentence “ treatments were performed in a controlled environment chamber under 16:8 hours light/dark photoperiod and 22 ℃ temperature conditions” and added the manufacturer of RhoChe thermocycler (LightCyclerR 480II, RhoChe, Basel, Switzerland); removed Table1 into the Supplementary Materials and added the names of the listed genes.
Subsection: Expand title from Immunoblotting to explain what was immunoblotted Line 90: “The SDS-PAGE gel concentration was 12.5%” is unnecessary. Include this information where logically belongs to, into the sentence in lines 81-82. For eg: Equal amounts of protein (30 ug) for each sample were subjected to SDS-PAGE, using a 12.5% gel. Explain the TBS abbreviation used here.
Answer: We changed it to “Equal amounts of protein (30 ug) for each sample were subjected to SDS-PAGE, using a 12.5% gel.” and explained TBS (NaCl, 8.8g; 2M,Tris-HCl, PH,7.6,5 mL and 995 mL H2O)
Subtitle needs to be reformulated. What the authors mean by assaying the nature of…? subsection needs references to the used procedures. Include details on the mechanical infection. How? Reference!
Answer: We explain it and add a Reference, as follows “To confirm the anti-APG8A polyclonal antibody specially recognizes ATG8 and ATG8-PE protein; a cabbage phospholipase D (PLD) was used to assay the nature of ATG8-PE[12]”.
Line 94: for consistency use 20% PEG only in the text.
Answer: 20% PEG was used throughout the text.
Expand subtitle to include what was imaged. Remove from line 116-117 : “The detailed method was a follows” – unnecessary. Reference the TEM method used.
Answer: We have removed the sentence “The detailed method was as follows” and added the reference [13]
Line 120-121: 0.1M phosphate buffer: sodium or potassium phosphate?
Answer: It’s “phosphate-buffered saline (PBS; 140 mM NaCl, 2.7 mM KCl, 10 mM Na2HPO4, 1.8 mM KH2PO4, pH 7.4)”
16 – line 133: ….”leaf samples grown under certain conditions were harvested at certain timepoints” – specify details instead of using “certain” Lines 135-137
Answer: We described it like this “200 mg of root and leaf samples grown under 20% PEG-8000 were harvested at certain time points (48, 96, 144, 192, 240 h).”
17– sentence “To visualize…” needs to be reformulated, is incorrect Line 137 – remove sentence
Answer: we have corrected it.
“Tunel staining was carried out…” - unnecessary Line 138 – specify xylene concentration
Answer: We have removed the sentence “Tunel staining was carried out…” and specified that “Samples were incubated in pure 2 changes of xylene”
Line 141 – “Proteinase K working solution”. Specify concentration/amounts used or units.
Answer: “Proteinase K working solution (20 ug Proteinase K powder distilled in 1 mL PBS)”
Lines 150-151 – Germany – spell with capital letters. Include details on the microscopy observations! Wavelengths, magnification, etc…or reference the used technique.
Answer: We have added some details as follows “The slides were then examined using a fluorescent microscope (Axiocam, Carl Zeiss, Germany) with 590 nm emission wavelength, 400×magnification.”
The same in 2.8 and 2.9! Include as 2.10 all details that pertain to Data analysis and statistics used!
Answer: we add a description pertaining to Data analysis and statistics used in
“Materials and method part” and described as follows
“Quantitative statistics
The protein bands on the membrane were analyzed using the Quantity One software (http://quantity-one.software.informer.com/). The data of qRT-PCR was analyzed by graphpad prism 5 software. All experimental data were then analyzed with the Student t test *, P < 0.05; **, P < 0.01; ***, P < 0.001. Throughout the study, the values are represented as the mean ± standard deviation of 3 independent.”
– uses % PEG consistently (not mixed with % g/mL PEG). Remove the word “chemical” from line 176, unnecessary Figure 1 description: consolidate panel descriptions to avoid redundancy.
For Eg. Seedling phenotypes (A), root length (B) and leaf length (C) following treatment with 10, 20 and 40% PEG-8000 for 48 hours…..Use this consolidation in all Figure descriptions.
Answer: we have changed according to reviewer’s suggestions and described like this:” Seedling phenotypes (A), root length (B) and leaf length (C) following treatment with 10, 20 and 40% PEG-8000 for 48 h. Seedling phenotypes (D), root length (E) and leaf length (F) following treatment with 20% PEG-8000 for 6, 24, 48 and 72 h.”
Figure 2. 20% PEG treatment at the 48 hours time point in roots on panel B (2.3 or 2.4) and F (1.8?) shows different relative mRNA levels, though those points overlap between approaches. How do the authors explain this discrepancy? If these data derive from different experiments, these differences (variations between individual samples) may be larger than the significant differences and the trend shown on panel F…
Answer: Yes, these data derived from different experiments, Firstly, different experiments cause some errors. Secondly, it is probably the sample of Figure 2F had a little degradation during the extraction of mRNA. We will repeat this experiment in the coming days.
Line 216 : “Since ATG8-PE is considered…” is unfinished sentence. Modify!
Answer: We have modified the sentence.
Figure 3, Lines 222-223: “Treatment with 20% PEG-8000 for 72 hours was used as a control”. Explain why!
Answer:The reason is described as follows “To confirm the anti-APG8A polyclonal antibody specially recognizes ATG8 and ATG8-PE protein…... ” and “Treatment with 20% PEG-8000 for 72 h was used as a control (because, ATG8 and ATG8-PE protein bands are obvious under this condition).”
subtitle and sentence in lines 247-248 needs to be reformulated, does not make sense:
Answer: We have reformulated the sentence and described like this “However, DNA ladder detection and tunel staining showed that the classical features of PCD (green nuclei stained by Tunel) were not observed in both roots and leaves of wheat seedlings (Figure 5C, and D). These data suggest that, the roots and leaves of wheat seedlings are survived when autophagy occurs.”
“….wheat seedlings are survival (survive?) when autophagy occurs”
Answer:We have changed “survival” to survived
Line 243: Remove “the following experiments were performed”. This is the results section, does not belong here.
Answer:We have changed it and described like this “To study whether the wheat seedlings undergo PCD or not when autophagy occurs, Tunel staining and DNA ladder analysis were performed.”
Figure 5 presents data from samples collected after 96 hours. Why this time-point and not some of the later or earlier points? The authors need to find a coherent manner to explain why do they present the array of data and why selected to present only some sets of data at a variety of different time-points during treatments.
Answer: Wheat seedlings did not present a wilting phenotype until 96 h-treatment; so, it is impossible that PCD occurs before 96 h.
Line 257 “ As a positive control plants were treated with 20% H2O2”. Why? These control treatments and their justification should be included into the appropriate sections of the Materials and Methods.
Answer: We added an explanation and described like this “As a positive control, plants were treated with 20% H2O2 for 96 h (Because high concentration of H2O2 results in PCD [14]; and this concentration is enough to induce PCD)”
Figure 6 – Are these panels representative examples of a larger set of data? (mentioned 3 experiments..) Than specify that these are representative examples and not averages.
Answer: Yes, these panels are representative examples of a larger set of data.
Discussion: The discussion is very superficial and does not concentrate on the evidence and data the paper presents, aligning data with the state of the art of current knowledge in the field of plant stress responses. The authors spend a disproportionately long section in the discussion in detailing potential mechanisms of drought induced autophagy and PCD, but their experiments did not investigate these mechanisms and did not show data in support of the described mechanisms. All this, at the expense of neglecting focusing on their own findings. I recommend re-writing and expanding the discussion to address the data specifically presented in the manuscript.
Answer: We have rewritten and expanded the discussion, and described as follows
Drought severely inhibits seedling growth [15]; often resulting in autophagy and PCD [16]. However, the relationship between autophagy and PCD remains unclear, as do the factors determining this relationship under drought stress. In this study, we revealed that autophagy precedes PCD in wheat seedlings during drought stress, with the duration of drought determining this relationship. The findings also revealed that wheat seedlings undergo autophagic survival, with subsequent PCD occurring at a much later stage, during more severe drought.
Drought caused by rapid global warming directly impacts agricultural productivity and poses a major challenge to the present-day agriculture [1]. Wheat is one of the most important crops worldwide [2], and in arid and semiarid areas, drought stress is the principal abiotic factor affecting wheat yield [3]. Therefore, it is significant to study the regulation mechnism of wheat on drought stress for improving drought resistance. Our study found that autophagy precedes PCD and inhibition of autophagy accelerated PCD under drought stress; which may improve wheat drought stress by enhancing autophagy ability appropriately.
3.1 PEG-8000 induces both autophagy and PCD in wheat seedlings
In Triticum dicoccoides, ATG8 protein and autophagy bodies are abundent in both the leaves and roots during drought stress[17]. Moreover, in common wheat, drought was found to increase the transcript levels of autophagy-related genes (ATGs) and induce certain proteins involved in the autophagy degradation pathway [18] . Drought stress was also found to induce PCD at a relatively early stage in the endosperm of winter wheat [19]. Therefore, both autophagy and PCD are induced by drought stress in wheat. In our study, using the common wheat cultivar Jimai 22, drought induced an increase in mRNA levels of ATG6 and ATG8, the formation of ATG8-PE, and the number of autophagic vacuoles in both a time- and concentration-dependent manner and induced PCD in time-dependent manner. Overall, we firstly established a molecular model of drought-induced autophagy and PCD in wheat seedlings, which will provide a standard for further study on the mechanism of autophagy and PCD under drought stress.
Additionaly, we use PEG-8000 simulated drought and in most experiments, 20% concentration was selected; because many existing methods for managing substrate water potential have employed PEG to simulate a specific water deficit exposure[4, 5]; and in wheat seedlings, the concentration of 20% PEG are often selected for related experiments [6-8]. Moreover, in our study, 20% PEG-8000 promoted autophagy just over 24h; and under this concentration, wheat seedlings are not be killed in a short time. The physiological responses to soil drought and PEG stress are similar in wheat, including lower leaf relative water content, lower water potential, increased osmotic regulation, and enhanced antioxidant capacity [9-11].
The mechanism of drought-induced autophagy and PCD may be related to the ABA signal pathway. Abscisic acid (ABA) is a plant hormone involved in stress responses, increasing significantly under drought stress [20]. Under stress, ABA triggers PYL (ABA receptor)-mediated activation of sucrose non-fermenting-related protein kinase 2 (SnRK2s), which phosphorylates the Target of Rapamycin (TOR) regulator Raptor, thereby inhibiting TOR activity [21]. As a result, inhibition of TOR activity promotes autophagy [22]. Moreover, ABA also induces ROS and H2O2 accumulation, contributing to plant PCD [23]. However, drought is also thought to induce autophagy via an ABA-independent pathway; for example, by binding the transcription factor ethylene response factor 5 (ERF5) to the promoters of ATG8d and ATG18h, or the heatshock transcription factor A1a (HsfA1a) to the promoters of ATG10 and ATG18f [21, 24].
3.2 Autophagy precedes PCD
In plants, autophagy is required for drought tolerance [16]; while under severe water stress, modification of root system architecture occurs due to PCD in the root meristem [25]. Understanding the relationship between PCD and autophagy during drought stress is therefore important in crops such as wheat. Although studies have shown the concomitant occurrence of autophagy and PCD in plants [26]; whether or not these two processes occur simultaneously remains unknown. In our study, we revealed for the first time that autophagy and PCD do not occur simultaneously in wheat seedlings under drought stress. That is, while autophagy occurred at a relatively early stage, PCD occurred much later, autophagy preceding PCD. The duration of drought stress was also found to determine occurrence, with short-term drought promoting autophagy and long-term drought promoting PCD. Prolonged drought stress is often associated with ROS accumulation since decreasing CO2 fixation is concomitant with increased electron leakage of triplet oxygen, which may eventually lead to PCD [27]. Moderate drought may produce low levels of ROS. Lower levels of ROS signalling positively induce autophagy activity, whereas higher ROS level would lead to rapid programmed cell death (PCD) [28]. In animals, the mechanism of autophagy and PCD also occurs sequentially, cleavage of ATG5 and ATG6 proteins mediating autophagy transformation to apoptosis [29, 30]. However, it is still unknown why autophagy precedes PCD under drought stress in plants. However, why autophagy precedes PCD in plants under drought stress remains unknown and requires further research in the future.
Based on above study, we can surmise that whether PCD or autophagy caused by abiotic stress may be related to the degree and duration time of stress. Moderate or short time stress may result in autophagy; and strong or long time stress result in PCD. The degree and duration time of stress may determine the switch from autophagy to PCD.
3.3 Autophagy is a cell survival process under drought stress.
As mentioned above, drought stress induces autophagy prior to PCD; however, the role of autophagy during drought stress remains unclear in common wheat seedlings [31]. Atg8-silenced wild emmer wheat plants were sensitive to drought stress in comparison to controls [17]. while activation of ATG genes promoted drought tolerance in tomato [32]. Overexpression of apple ATG18a in tomato and apple plants was also found to increase resistance to drought compared to wild-type plants [33]. suggesting that autophagy plays a pro-survival role. Meanwhile, in ipomoea petals, the autophagy inhibitor 3-MA was found to accelerate visual senescence [34] , and similarly, our study revealed an increase in drought-induced PCD following 3-MA treatment and the knocking down of ATG6, respectively. These findings therefore suggest that autophagy also plays a cell survival role in wheat seedlings under drought stress. Regarding the mechanism of autophagy-induced survival under drought stress, some studies suggest that drought stress leads to accumulation of ROS and H2O2, which are then possibly removed via the autophagic response, preventing the induction of PCD. Regarding the mechanism of autophagy-induced survival under drought stress, some studies suggest that drought stress leads to accumulation of ROS and H2O2, which are then possibly removed via the autophagic response, preventing the induction of PCD [28, 35]. However, clarification of this mechanism is required.
Line 323 “In plants” is repeated. Lines 352 and 353 – “however” is used in two consecutive sentences- redundant. Reformulate!
Answer: we have deleted it
Answer:we have reformulated it
Figure 9 does not add specific value to the manuscript. It is clear already from the data that autophagy precedes PCD and PCD will not occur until the late, sever phases of stress. Either remove Figure 9, or expand it and tie in additional details on the hypothesized molecular / cellular mechanisms and details behind autophagy and PCD based on information available in the literature to integrate the novel information, as also discussed/hypothesized in the Discussion.
Answer: we have removed Figure 9
Author contributions: “…analyzed the experiments..”, maybe mean “analyzed the data…”?
Answer: Yes, it is “analyzed the data…”
Reviewer 3 Report
The manuscript is interesting for its problems. The authors wrote it relatively carefully. The abstract is corresponding, rather general. The methodology is clearly described. I have some comments on this section. PEG establishes water stress, but from a physiological point of view, it is more of an osmotic stress. As a measure of stress I would mention the water potential of plants or substrate and RWC. Thus it is not entirely clear whether the plants were actually under stress. Please add statistical methods. The graphic part is on a good level. I would also mention values and differences in the results. The discussion is rather descriptive. I really recommend discussing the results obtained.
Author Response
Dear reviewer3
Thank you for your advice; the answers are as follows:
Open Review
(x) I would not like to sign my review report
( ) I would like to sign my review report
English language and style
( ) Extensive editing of English language and style required
( ) Moderate English changes required
( ) English language and style are fine/minor spell check required
(x) I don't feel qualified to judge about the English language and style
Yes Can be improved Must be improved Not applicable
Does the introduction provide sufficient background and include all relevant references? (x) ( ) ( ) ( )
Is the research design appropriate? ( ) (x) ( ) ( )
Are the methods adequately described? ( ) (x) ( ) ( )
Are the results clearly presented? ( ) (x) ( ) ( )
Are the conclusions supported by the results? (x) ( ) ( ) ( )
Comments and Suggestions for Authors
The manuscript is interesting for its problems. The authors wrote it relatively carefully. The abstract is corresponding, rather general. The methodology is clearly described. I have some comments on this section. PEG establishes water stress, but from a physiological point of view, it is more of an osmotic stress. As a measure of stress I would mention the water potential of plants or substrate and RWC. Thus it is not entirely clear whether the plants were actually under stress. Please add statistical methods. The graphic part is on a good level. I would also mention values and differences in the results. The discussion is rather descriptive. I really recommend discussing the results obtained.
PEG establishes water stress, but from a physiological point of view, it is more of an osmotic stress. As a measure of stress I would mention the water potential of plants or substrate and RWC. Thus it is not entirely clear whether the plants were actually under stress.
Answer: we added some explanations about the use of PEG, described as follows
“Additionaly, we use PEG-8000 simulated drought and in most experiments, 20% concentration was selected;because many existing methods for managing substrate water potential have employed PEG to simulate a specific water deficit exposure[4, 5];and in wheat seedlings,the concentration of 20% PEG are often selected for related experiments [6-8]. The physiological responses to soil drought and PEG stress are similar in wheat, including lower leaf relative water content, lower water potential, increased osmotic regulation, and enhanced antioxidant capacity [9-11]”.
Please add statistical methods
Answer: We have added statistical methods in“Materials and method part” and described as follows
“Quantitative statistics
The protein bands on the membrane were analyzed using the Quantity One software (http://quantity-one.software.informer.com/). The data of qRT-PCR was analyzed by graphpad prism 5 software. All experimental data were then analyzed with the Student t test *, P < 0.05; **, P < 0.01; ***, P < 0.001. Throughout the study, the values are represented as the mean ± standard deviation of 3 independent.”
The discussion is rather descriptive. I really recommend discussing the results obtained.
Answer: We have rewritten and expanded the discussion, and described as follows
Drought severely inhibits seedling growth [15]; often resulting in autophagy and PCD [16]. However, the relationship between autophagy and PCD remains unclear, as do the factors determining this relationship under drought stress. In this study, we revealed that autophagy precedes PCD in wheat seedlings during drought stress, with the duration of drought determining this relationship. The findings also revealed that wheat seedlings undergo autophagic survival, with subsequent PCD occurring at a much later stage, during more severe drought.
Drought caused by rapid global warming directly impacts agricultural productivity and poses a major challenge to the present-day agriculture [1]. Wheat is one of the most important crops worldwide [2], and in arid and semiarid areas, drought stress is the principal abiotic factor affecting wheat yield [3]. Therefore, it is significant to study the regulation mechnism of wheat on drought stress for improving drought resistance. Our study found that autophagy precedes PCD and inhibition of autophagy accelerated PCD under drought stress; which may improve wheat drought stress by enhancing autophagy ability appropriately.
3.1 PEG-8000 induces both autophagy and PCD in wheat seedlings
In Triticum dicoccoides, ATG8 protein and autophagy bodies are abundent in both the leaves and roots during drought stress[17]. Moreover, in common wheat, drought was found to increase the transcript levels of autophagy-related genes (ATGs) and induce certain proteins involved in the autophagy degradation pathway [18] . Drought stress was also found to induce PCD at a relatively early stage in the endosperm of winter wheat [19]. Therefore, both autophagy and PCD are induced by drought stress in wheat. In our study, using the common wheat cultivar Jimai 22, drought induced an increase in mRNA levels of ATG6 and ATG8, the formation of ATG8-PE, and the number of autophagic vacuoles in both a time- and concentration-dependent manner and induced PCD in time-dependent manner. Overall, we firstly established a molecular model of drought-induced autophagy and PCD in wheat seedlings, which will provide a standard for further study on the mechanism of autophagy and PCD under drought stress.
Additionaly, we use PEG-8000 simulated drought and in most experiments, 20% concentration was selected; because many existing methods for managing substrate water potential have employed PEG to simulate a specific water deficit exposure[4, 5]; and in wheat seedlings, the concentration of 20% PEG are often selected for related experiments [6-8]. Moreover, in our study, 20% PEG-8000 promoted autophagy just over 24h; and under this concentration, wheat seedlings are not be killed in a short time. The physiological responses to soil drought and PEG stress are similar in wheat, including lower leaf relative water content, lower water potential, increased osmotic regulation, and enhanced antioxidant capacity [9-11].
The mechanism of drought-induced autophagy and PCD may be related to the ABA signal pathway. Abscisic acid (ABA) is a plant hormone involved in stress responses, increasing significantly under drought stress [20]. Under stress, ABA triggers PYL (ABA receptor)-mediated activation of sucrose non-fermenting-related protein kinase 2 (SnRK2s), which phosphorylates the Target of Rapamycin (TOR) regulator Raptor, thereby inhibiting TOR activity [21]. As a result, inhibition of TOR activity promotes autophagy [22]. Moreover, ABA also induces ROS and H2O2 accumulation, contributing to plant PCD [23]. However, drought is also thought to induce autophagy via an ABA-independent pathway; for example, by binding the transcription factor ethylene response factor 5 (ERF5) to the promoters of ATG8d and ATG18h, or the heatshock transcription factor A1a (HsfA1a) to the promoters of ATG10 and ATG18f [21, 24].
3.2 Autophagy precedes PCD
In plants, autophagy is required for drought tolerance [16]; while under severe water stress, modification of root system architecture occurs due to PCD in the root meristem [25]. Understanding the relationship between PCD and autophagy during drought stress is therefore important in crops such as wheat. Although studies have shown the concomitant occurrence of autophagy and PCD in plants [26]; whether or not these two processes occur simultaneously remains unknown. In our study, we revealed for the first time that autophagy and PCD do not occur simultaneously in wheat seedlings under drought stress. That is, while autophagy occurred at a relatively early stage, PCD occurred much later, autophagy preceding PCD. The duration of drought stress was also found to determine occurrence, with short-term drought promoting autophagy and long-term drought promoting PCD. Prolonged drought stress is often associated with ROS accumulation since decreasing CO2 fixation is concomitant with increased electron leakage of triplet oxygen, which may eventually lead to PCD [27]. Moderate drought may produce low levels of ROS. Lower levels of ROS signalling positively induce autophagy activity, whereas higher ROS level would lead to rapid programmed cell death (PCD) [28]. In animals, the mechanism of autophagy and PCD also occurs sequentially, cleavage of ATG5 and ATG6 proteins mediating autophagy transformation to apoptosis [29, 30]. However, it is still unknown why autophagy precedes PCD under drought stress in plants. However, why autophagy precedes PCD in plants under drought stress remains unknown and requires further research in the future.
Based on above study, we can surmise that whether PCD or autophagy caused by abiotic stress may be related to the degree and duration time of stress. Moderate or short time stress may result in autophagy; and strong or long time stress result in PCD. The degree and duration time of stress may determine the switch from autophagy to PCD.
3.3 Autophagy is a cell survival process under drought stress.
As mentioned above, drought stress induces autophagy prior to PCD; however, the role of autophagy during drought stress remains unclear in common wheat seedlings [31]. Atg8-silenced wild emmer wheat plants were sensitive to drought stress in comparison to controls [17]. while activation of ATG genes promoted drought tolerance in tomato [32]. Overexpression of apple ATG18a in tomato and apple plants was also found to increase resistance to drought compared to wild-type plants [33]. suggesting that autophagy plays a pro-survival role. Meanwhile, in ipomoea petals, the autophagy inhibitor 3-MA was found to accelerate visual senescence [34] , and similarly, our study revealed an increase in drought-induced PCD following 3-MA treatment and the knocking down of ATG6, respectively. These findings therefore suggest that autophagy also plays a cell survival role in wheat seedlings under drought stress. Regarding the mechanism of autophagy-induced survival under drought stress, some studies suggest that drought stress leads to accumulation of ROS and H2O2, which are then possibly removed via the autophagic response, preventing the induction of PCD. Regarding the mechanism of autophagy-induced survival under drought stress, some studies suggest that drought stress leads to accumulation of ROS and H2O2, which are then possibly removed via the autophagic response, preventing the induction of PCD [28, 35]. However, clarification of this mechanism is required.

Round 2
Reviewer 2 Report
The authors of this revised manuscript addressed all questions and concerns I had with the study, significantly improving the manuscript.
I solely have one recommendation left that is to perform a thorough read-though of the manuscript to correct typos, errors in the English formulation and catch some of the textual inconsistencies which still appear in the text at many instances, especially in the newly added sections. A read-through with the copy-editor or a native English speaker would be of benefit.
Here are a few (not all) examples:
“After germination for one week, Pick out seedlings that grow consistently.” replace with “After germination for one week, seedlings of consistent growth were selected for experiments” “Moreover, in our study, 20% PEG-8000 promoted autophagy just over 24h; and under this concentration, wheat seedlings are not be killed in a short time. The physiological responses to soil drought and PEG stress are similar in wheat, including lower leaf relative water content, lower water potential, increased osmotic regulation, and enhanced antioxidant capacity [9-11].” Rephrase “Moreover, in our study, 20% PEG-8000 promoted autophagy later than 24h, therefore wheat seedlings were not killed in a short time. The physiological responses to soil drought and PEG-induced osmotic stress are similar in wheat, including decreased (lowered?) leaf relative water content, water potential, increased osmotic regulation, and enhanced antioxidant capacity [9-11].” “The first leaf and about 5 cm from root tip were sampled during the treatment. The samples can be stored in a -70℃ refrigerator for the following Quantitative real-time reverse transcription-PCR (qRT-PCR) and Western blot analysis. However, fresh samples must be provided for Transmission electron microscopy (TEM) and PCD detection.” Reformulate and correct: “The first leaf and about 5 cm from the root tip were sampled during the treatment. The samples were stored in a -70℃ refrigerator for quantitative real-time reverse transcription-PCR (qRT-PCR) and Western blot analysis. However, for Transmission electron microscopy (TEM) and PCD detection fresh samples were used.” Figure 2. (line 85) “Effect of drought on ATG6 and ATG8 expression. Concentration effect of 10, 20, and 40% PEG-8000 on root mRNA levels ….” – when using x% PEG is obvious that is a concentration value. Also, since the experiment was an osmotic stress experiment simulating drought via the use of PEG, simplify and specify: “Effect of PEG-induced stress simulating drought on ATG6 and ATG8 expression. The effect of 10, 20, and 40% PEG-8000 on root mRNA levels ….” Line 413-414: “Throughout the study, the values are represented as the mean ± standard deviation of 3 independent.” This sentence is incomplete. ….”3 independent samples (experiments/treatments…)”Author Response
Dear reviewer
We have revised the manuscript according to the reviewer’s suggestions. The modified part is marked with blue font in manuscript. Moreover, the manuscript has been edited by MDPI. The answers to the reviewer are as follows.
The authors of this revised manuscript addressed all questions and concerns I had with the study, significantly improving the manuscript.
I solely have one recommendation left that is to perform a thorough read-though of the manuscript to correct typos, errors in the English formulation and catch some of the textual inconsistencies which still appear in the text at many instances, especially in the newly added sections. A read-through with the copy-editor or a native English speaker would be of benefit.
Here are a few (not all) examples:
“After germination for one week, Pick out seedlings that grow consistently.” replace with “After germination for one week, seedlings of consistent growth were selected for experiments”Answer:We have replaced it with the sentence “After germination for one week, seedlings of consistent growth were selected for experiments”
“Moreover, in our study, 20% PEG-8000 promoted autophagy just over 24h; and under this concentration, wheat seedlings are not be killed in a short time. The physiological responses to soil drought and PEG stress are similar in wheat, including lower leaf relative water content, lower water potential, increased osmotic regulation, and enhanced antioxidant capacity [9-11].” Rephrase “Moreover, in our study, 20% PEG-8000 promoted autophagy later than 24h, therefore wheat seedlings were not killed in a short time. The physiological responses to soil drought and PEG-induced osmotic stress are similar in wheat, including decreased (lowered?) leaf relative water content, water potential, increased osmotic regulation, and enhanced antioxidant capacity [9-11].”
Answer:We have rephrased it.
“The first leaf and about 5 cm from root tip were sampled during the treatment. The samples can be stored in a -70℃ refrigerator for the following Quantitative real-time reverse transcription-PCR (qRT-PCR) and Western blot analysis. However, fresh samples must be provided for Transmission electron microscopy (TEM) and PCD detection.” Reformulate and correct: “The first leaf and about 5 cm from the root tip were sampled during the treatment. The samples were stored in a -70℃ refrigerator for quantitative real-time reverse transcription-PCR (qRT-PCR) and Western blot analysis. However, for Transmission electron microscopy (TEM) and PCD detection fresh samples were used.”
Answer:We have rephrased it.
Figure 2. (line 85) “Effect of drought on ATG6 and ATG8 expression. Concentration effect of 10, 20, and 40% PEG-8000 on root mRNA levels ….” – when using x% PEG is obvious that is a concentration value. Also, since the experiment was an osmotic stress experiment simulating drought via the use of PEG, simplify and specify: “Effect of PEG-induced stress simulating drought on ATG6 and ATG8 expression. The effect of 10, 20, and 40% PEG-8000 on root mRNA levels ….”
Answer: we have simplified and specified it.
Line 413-414: “Throughout the study, the values are represented as the mean ± standard deviation of 3 independent.” This sentence is incomplete. ….”3 independent samples (experiments/treatments…)”
Answer: we have corrected it.

This manuscript is a resubmission of an earlier submission. The following is a list of the peer review reports and author responses from that submission.
Round 1
Reviewer 1 Report
In this article, Li and collaborators evaluate the relationship between autophagy and programmed cell death under drought in wheat seedlings. It was previously known that autophagy is a survival mechanism for plants under drought and other biotic stresses, and this was observed not only in arabidopsis but also in wheat. The key contribution of this work is the time-course analysis of the drought response and the evidence that autophagy precedes PCD. However, the experimental design is not conducive to an insightful mechanism for this process. Indeed, the authors discuss the possible role of accumulation of reactive oxygen species as the triggers of PCD after autophagy, but there were no experiments that allowed for the testing of the hypothesis. In this regard, recent work in tomato identified the mitochondrial alternative oxidase (AOX) as a regulator of autophagy during drought response.
It is my opinion that this work is still incomplete in the sense that it provides little to the scientific community regarding the very important topic that it addresses. I believe that a few more experiments testing some of the sound hypothesis addressed in the discussion would go a long way in improving the relevance of this work.
On the other hand, the English needs to be revised throughout the paper are there are very many serious mistakes that make it difficult to read.
Reviewer 2 Report
Reviewer's Report for manuscript entitled: “Wheat seedlings undergoes autophagic survival and subsequent programmed cell death under drought stress”
Ref. ijms-556921
Although the manuscript examines an interesting subject and the experimental procedure is executed with accuracy, and meaningful results were produced, it cannot be published in the present form.
English language is poorly used. The manuscript contains numerous language and spelling errors, often making text comprehension challenging. I strongly recommend the manuscript being revised by an English language professional, so several parts become clarified.
Moreover, this manuscript presents several major shortcomings:
1. The abstract is poorly written merely listing results. It would be better to summarize introduction, hypotheses, results, conclusion..
2. The introduction section is very poor. Purposes of study are not stated clearly, and background literature is insufficiently reviewed. I believe that the authors wrongly state that the “The study provides new insights into the regulation mechanism of autophagy”. Many review articles exist dealing with autophagy and drought stress (e.g doi:10.1098/rsob.180162, https://doi.org/10.1111/pce.13404), that are not properly cited. The authors also use terminology wrongly. For example, lines 40-45, the authors state that plant cells contain lysomes. That is not true, plants do not have lysomes and the vacuole of the plant cell is the key organelle orchestrating PCD processes.
3. The materials and methods section does not describe sufficiently the methodological approach. Number of plants collected, experiment replication, and fixation process are unclear.
4. The Results section presents numerous inconsistencies. Even though results are meant to be predominantly descriptive, the rational about the choice of each parameter measured needs to be briefly explained. The distinction between statistically significant and non significant results is not clear, so presence of a trend cannot be determined. Additionally, authors need to clearly state which plant roots parameters they studied. Moreover the authors should clarify which of the root parameters they studied. Moreover, electron micrographs included are of low quality, and do not support conclusions in the present form.
TEM figures presented are only from leaves while the authors have also studied roots. I would recommend also root TEM figures to be also shown. How sure the authors are that the TEM figures presented are not simply plasmolysed cells due to the fixation process?
In figure 5, the positive control of PI staining fails to stain the nuclei, and seem to stain the whole cell. I would recommend replacing the PI staining protocol to check cell vitality with the acridine orange (AO) and propidium iodide (PI) protocol.
5. Discussion needs to be revised to match results. It fails to discuss the scope of the manuscript and should be re-written to focus concisely.